# Epigenetic and Epitranscriptomic Gene Regulation in *Plasmodium falciparum* and How We Can Use It against Malaria

**DOI:** 10.3390/genes13101734

**Published:** 2022-09-27

**Authors:** Rafael Serrano-Durán, Diana López-Farfán, Elena Gómez-Díaz

**Affiliations:** Instituto de Parasitología y Biomedicina López Neyra (IPBLN-CSIC), 18016 Granada, Spain

**Keywords:** chromatin structure, transcriptional regulation, *Plasmodium falciparum*, mosquito, cellular plasticity

## Abstract

Malaria, caused by *Plasmodium* parasites, is still one of the biggest global health challenges. *P. falciparum* is the deadliest species to humans. In this review, we discuss how this parasite develops and adapts to the complex and heterogenous environments of its two hosts thanks to varied chromatin-associated and epigenetic mechanisms. First, one small family of transcription factors, the ApiAP2 proteins, functions as master regulators of spatio-temporal patterns of gene expression through the parasite life cycle. In addition, chromatin plasticity determines variable parasite cell phenotypes that link to parasite growth, virulence and transmission, enabling parasite adaptation within host conditions. In recent years, epitranscriptomics is emerging as a new regulatory layer of gene expression. We present evidence of the variety of tRNA and mRNA modifications that are being characterized in *Plasmodium* spp., and the dynamic changes in their abundance during parasite development and cell fate. We end up outlining that new biological systems, like the mosquito model, to decipher the unknowns about epigenetic mechanisms in vivo; and novel methodologies, to study the function of RNA modifications; are needed to discover the Achilles heel of the parasite. With this new knowledge, future strategies manipulating the epigenetics and epitranscriptomic machinery of the parasite have the potential of providing new weapons against malaria.

## 1. Introduction

Despite being one of the first recorded illness in history [1], malaria is still one of the main health burdens in the world, threatening half of the global population. The most recent records show that the global pandemic of COVID-19 has aggravated the problem, with more than 240 million cases and 600 thousand deaths worldwide [2]. Malaria is endemic in 85 countries, being most prevalent in West and Sub-Saharan Africa. [2].

The disease is caused by *Apicomplexan* parasites of the genus *Plasmodium* that infect different mammalian hosts. *P. falciparum* is the predominant human malaria species, causing the most severe symptomatology and the highest mortality rate [2].

Once *P. falciparum* enters our bloodstream, by means of an infected female *Anopheles* mosquito bite, the sporozoites invade liver hepatocytes where they multiply for two weeks. Thousands of merozoites per liver cell are released, and then re-enter the blood stream and invade the erythrocytes. Then, each parasite cell enters a continuous cycle called the intraerythrocytic developmental cycle (IDC), developing into ring, trophozoite and the schizont stage. After the formation of a mature schizont, several merozoites tear out the erythrocyte and reinfect new red blood cells. Some of these cycling parasites, in response to certain conditions within the host [3], differentiate into transmissible sexual forms: the gametocytes. This blood cycle takes approximately 48 h.

The life cycle in the mosquito starts when a female *Anopheles* bites an infected human, taking a blood meal with gametocytes that reaches the mosquito’s midgut, where they become activated and form extracellular gametes induced by changes in temperature, pH and xanthurenic acid [4]. It is in the mosquito when the sexual reproduction of the parasite take place: the macrogametes are fertilized by microgametes, forming diploid zygotes. The zygote develops into a motile ookinete that traverses the midgut epithelium and becomes an oocyst. Inside the oocyst, the parasite divides by mitosis forming multiple (hundreds) transmissible forms called sporozoites that tear out the oocysts when mature. By day 14 approximately, the parasites migrate from the mosquito midgut to the salivary glands, where they can be transmitted and infect another human host in another mosquito bite, closing the parasite life cycle.

To survive in such a complex life cycle involving two hosts, intra- and extra-cellular life forms and multiple host tissues, the parasite requires different sets of proteins and a very tight control of cell cycle progression. For such, transcription is precisely regulated in space and time by a relatively small repertoire of transcription factors called ApiAP2 proteins [5,6]. How about thirty proteins cooperate and, in some cases compete, to spatiotemporally regulate thousands of parasite genes, is a subject of intense investigation. In addition, changes in chromatin structure impact transcription, playing a key role in malaria parasites developmental progression. Each life cycle stage has its unique chromatin signature, that associates with state-specific transcriptional states and prime the parasite for the next developmental transition [7,8,9]. Various chromatin-associated processes orchestrate these changes: histone post-translational modifications and histone variants, nucleosome landscape, 3D chromatin organization, as well as non-coding RNA (ncRNA).

Apart of a very complex life cycle, the most striking and deadly feature of the malaria parasite is its ability to adapt rapidly and reversibly, without changing its genome, to the variable environment of the host. This is achieved thanks to cell plasticity, the generation of alternative phenotypes in the parasite population [10]. And this plasticity is what makes *Plasmodium* epigenetics shine. Epigenetics is linked to almost every biological process of *P. falciparum*, controlling a variety of survival strategies such as immune system evasion, transmission and drug resistance.

In this review we will deepen into chromatin and epigenetic mechanisms of gene regulation, mainly AP2 transcription factors, histone post-translational modifications and nuclear organisations, that control *P. falciparum* cell fate and plasticity. In addition, we will discuss new insights into the role of epitranscriptomics, also named “RNA epigenetics”, as a new layer of gene regulation in parasite development and adaptation. We will end up envisioning future strategies of manipulating the epigenetic and epitranscriptomic machinery of the parasite for malaria eradication.

## 2. Chromatin Regulation of Parasite Development

Fundamental chromatin organization regulation is conserved in *Plasmodium* parasites, sharing the main principles with their eukaryotic relatives, as well as the repertoire of histone post translational modifications [11,12,13]. One distinctive feature of the *P. falciparum* genome that impacts gene expression regulation is its elevated AT content. Indeed, it has one of the highest in all eukaryotes with an average rate of 80%, that surpass the 90% in introns and intergenic regions [14]. Long AT-rich segments have been suggested to impede nucleosome formation [15]. However, the presence of species-specific histone variants like PfH2B.Z and PfH2A.Z, which localise preferentially in AT-rich regions, have been proposed as the mechanism that would favour chromatin compactness [15].

During parasite development, chromatin changes dynamically, shaping parasite cell fate (Figure 1). Most of the parasite chromatin is in a stable euchromatin state which is determined by the DNA sequence that is bound by specific regulatory proteins [16]. These dynamic chromatin changes depend on a highly coordinated regulatory cascade [17,18]. Stage-specific transcription factors are required at different times and in different host tissues and cells to activate or repress gene expression. However, *P. falciparum* is notable for having one of the lowest ratios of predicted specific transcription factors versus total number of genes in eukaryotes [19]. Different hypotheses may explain this apparent lack of transcription factors. First, sequence redundancy of the parasite genome may allow the same TF to bind and regulate multiple promoters [6]. Another explanation is the temporal dynamics of gene transcription which in *Plasmodium* follows a “just-in-time” pattern [17]. That is, developmental and cell-fate determining TFs are temporally expressed and activate the expression of multiple stage-specific genes. Finally, the multiplicity of regulatory feedback loops that can be positive, negative and autoregulatory, involving other TFs as all well as long non-coding RNAs (lncRNAs) [7,20,21,22,23]. The only family of transcription factors in *Plasmodium*: the ApiAP2 family of DNA-binding proteins, comprises 27 members in *P. falciparum*. These AP2 transcription factors are specific of apicomplexan parasites and analogous to the plant Apetela2/Ethylene Response Factor (AP2/ERF) DNA-binding proteins [5,6]. Despite being so few, they act as master regulators controlling the expression of hundreds of genes at specific life stages [24]. This is the case for an example of the master regulators AP2-G and AP2-G2 in gametocytes [20,25], AP2-O in ookinetes [26], AP2-SP in sporozoites [27] and AP2-L in liver stages [28]. These AP2 proteins have been shown to act cooperatively activating transcription, although they are a few that act as repressors [7]. They are developmentally regulated, and their expression is usually under the control of the previous stage-controlling AP2 and reinforced by positive feedback loops [20,21,22]. Therefore, each transcriptional state prepares the background for the development of the next stage.

### 2.1. Chromatin Regulation during Intraerythocytic Development

Almost all we know about chromatin regulation in *Plasmodium* come from studies on the Intraerythrocytic Developmental Cycle (IDC). It is well described that the transcriptome and the epigenome of *P. falciparum* during the IDC in the human is highly dynamic. The transcriptional start sites of stage-active genes are marked with a combination of histone H3 acetylations H3K18ac and H3K27ac, that associates with an enrichment of AP2-I and bromodomain protein 1 (BDP1) reader protein in those loci [29]. The ring state is characterized by its low transcriptional activity [17,18]. This might be due to the enrichment of nucleosomes and the global compaction of the nucleus that have been shown in this state [8,9,30]. In addition to the general condensation, the number and size of nuclear pores is reduced [31]. The euchromatin in the ring stage is marked by the increased abundance of histone H4 modifications H4K20me1 and H4K8ac [11,32]. When the parasite progresses to trophozoite, it undergoes important morphological changes. In order to grow and form the daughter merozoites, the transcription is globally activated [17,18], with a more open chromatin, increasing the genome volume [8]. The promoter regions become depleted of nucleosomes [9,30,33], and the histones in the promoters of active genes become enriched in activation marks such as H3K9ac, H3K14ac and H3K4me3 [13,32]. On the other hand, genes that are not transcribed at this stage, like sexual and mosquito stages genes, show high nucleosome occupancy and heterochromatic marks like H3K9me3 [34,35,36,37]. In correlation with the active state of transcription of the trophozoite, the nuclear pores increase in number, especially in transcriptionally active domains in order to assist RNA export [31,38]. Finally, in the schizont stage the genome is condensed again, and in order to form the merozoites, nucleosomes are repacked [8,9,30]. As a result, the transcription rate drops, and the nuclear pores decrease in number [31]. The AP2-I transcription factor and PfBDP1 specially control the schizont transcriptional program, activating invasion genes. The promoter regions of these active genes are marked with acetylated histones [18,39,40,41]. The schizonts have a particular histone post-translational modification (PTM) landscape, enriched in unique activation marks such as H3K4me2, H3K4me3&ac, H3.3K9me1, H3K122ac and H4K16a [13], contrasting with a more reduced global transcription.

Taken together, available evidence shows that the development of the malaria parasite within the erythrocyte is highly regulated at the level of chromatin structure. Whether these patterns and mechanisms are similar in other stages of parasite development, requires further investigation.

### 2.2. The Knowns and the Unknowns about Chromatin Regulation in the Mosquito

Mosquito stages remain poorly investigated. This is in part due to technical limitations. The bottleneck in the parasite population following transmission results in very low parasite numbers in the invertebrate host, which makes it difficult to isolate the different mosquito stages at sufficient purity and quantity. Moreover, multiple infections, consisting of multiple strains and/or genotypes, often co-occur in the mosquito resulting in asynchronous and heterogenous parasite populations that hinder bulk transcriptomic and epigenomic analysis. As a consequence, the study chromatin regulation in this part of the parasite life cycle still remains a challenge.

The few studies that analyse chromatin regulation in mosquito stages, point to dynamic changes in chromatin states that correlate with stage-specific transcriptional programs [22,42,43]. However, data is still scarce for most part of the sporogonic development. That is, the chromatin landscape evolution from the gametocyte until the formation of the oocysts, is largely unknown. Furthermore, the mosquito life cycle is epigenetically unique because of the process of meiosis. It is known that an epigenetic reprograming during meiosis is common in animals [44], however it has not been studied in *Plasmodium*. We know that meiosis occurs in the diploid zygote stage and that the parasite possesses the conserved enzymatic machinery (Dmc1 and Rad51) [45]. However, it is currently unknown whether there is a global erase of epigenetic marks following recombination, or some type of epigenetic memory is retained [46]. After meiotic division, the mature ookinete crosses the midgut epithelium, evolving into an oocyte, a slightly more well-studied stage. At this point, the transcription of genes involved in growth, metabolism, transcription and splicing is activated, and this expression is linked to enrichment in H3K9ac and H3K27ac [22,43]. In addition, AP2-O act as a master regulator in oocysts where it has been shown to bind in promoter regions of more than 500 genes [47]. Indeed, AP2-O and AP2-O2 have been shown to be essential for development in *P. falciparum* and rodent parasites [26,47,48,49]. In sporozoites, once released from the oocysts, the heterochromatin domains are expanded beyond subtelomeric regions, silencing invasion and virulence genes [50]. This heterochromatin is marked with the usually repressive H3K9me3 modification; however, some genes appear in a bivalent state and continue being moderately expressed despite having this mark [22]. On the other hand, activating cell traversal and hepatocyte invasion genes display H3K9ac, H3K27ac and H3K4me3 and long-range interactions along the chromosomes [22,50,51,52]. Although the roles of ApiAP2 transcription factors in sporogony are not fully understood, studies in rodent sporozoites show that AP2-SP is bound to sporozoite-specific gene promotes, being indispensable for sporozoite development [27]. The ortholog in *P. falciparum*, AP2-EXP (PF3D7_1466400) has its maximum expression in sporozoites, and displays binding sites in the promoters of hundreds of sporozoite-specific genes, pointing to be the master regulator of this stage of development [22]. In addition to AP2-SP, AP2-SP2 and AP2-SP3 seem to be essential in sporozoite development [48,49].

Several studies have shown that most of the transcripts expressed in sporozoites are not translated into proteins [42,53,54]. This translational regulation is manifested by two pathways: overall inhibition of translation and transcript-specific silencing. The global inhibition is achieved though phosphorylation of the translation initiation factor, eIF2α [55]. It is known that specific transcript-silencing operates in sporozoites, an example is UIS4, one of the most abundant mRNA in this stage, but whose protein is not synthesized at this stage [53,54]; nevertheless, the mechanisms behind this silencing are not known. As we will see, gametocytes have a similar translational inhibition in which epitranscriptomic mechanisms could be involved [56,57,58]. Maybe, epitranscriptomics could be playing a sim6ilar role in the other transmissible stage, the sporozoite.

In sum, compared to IDC stages, data about chromatin accessibility, AP2 binding and 3D chromatin structure controlling mosquito stages developmental transitions is mostly lacking. Deciphering the many unknowns in the development regulation in the mosquito stages represents a vein with great potential for further studies.

## 3. Epigenetic Mechanisms Controlling Parasite Adaptation

Epigenetic regulation plays a crucial role in the cell plasticity of the malaria parasite. This plasticity is a survival strategy shared by many pathogens that allows the parasite to overcome the unpredictable and changing environments within the human and mosquito hosts. In isogenic populations, different phenotypes can arise thanks to epigenetic regulation. The genes whose expression can change between different parasite cells of the same genotype/clone are referred as Clonally Variant Genes (CVGs). These could be found either in an active or repressed state in genetically indistinguishable parasites. These epigenetic states can be inherited. Phenotypic variation can arise by two different processes. Bet-hedging, where transcriptional variants arise a priori, and the new environments act as selective sieve sorting; and adaptive plasticity, where the new phenotypic variants arise as directed transcriptional responses to an environmental change [10]. Evidence of either strategy exist in the blood cycle and in-vitro conditions [23,59,60,61,62,63,64,65,66], although the ability of malaria parasites to actively react to fluctuating conditions was considered controversial a few years ago [67,68].

Given that the CVG activation state does not depend on its primary DNA sequence, the basic regulatory mechanisms that control Clonally Variant Gene Expression (CVGE) is truly epigenetic and relies on facultative heterochromatin. This means that the trait is heritable, being passed through multiple rounds of replication [69]. It is also transitory so the activation/repressive state can switch with low frequency [70,71], developing heterogeneity in response to external signals or even in a uniform environment, in a stochastic manner. The mechanisms of arbitrary switching are not well studied, but two molecular mechanisms have been proposed: epigenetic regulator’s level changes and transmission errors of the epigenetic memory [10]. The chromatin state of CVGs can be reshaped by different epigenetic mechanisms. The state of H3K9 near the transcriptional start site is acetylated in active CVGs and tri-methylated in silenced CVGs [16,72,73,74,75,76,77]. The differences in expression are also determined by the localization of the locus within the nucleus [76,78,79,80,81]. It is also known that the H3K9me3 mark of silenced CVGs interact with Heterochromatin Protein 1 (HP1), that condensates the chromatin and arrest the area to the nuclear periphery, silencing the locus [82,83]. As we shall see below, there are other mechanisms and enzymes that control specific CVGE families. Whether these are generalizable to other CVG families and/or life-stages, has yet to be confirmed. In addition, there might be further regulatory proteins and CVGs that remain to be discovered, for example in the less studied mosquito stages.

Hereinafter, several examples of *Plasmodium* adaptation strategies involving CVGE that are under epigenetic control are explained. Amongst these, the hallmark examples are antigenic variation (bet-hedging strategy) and sexual differentiation (adaptation plasticity). All what we describe occur linked to the blood cycle in the human and in vitro conditions and have used *P. falciparum* laboratory reference strains.

### 3.1. Antigenic Variation and Immune Evasion

*P. falciparum* genome encodes different gene families of antigens like *var*, *Pfmc-2TM*, *rifin* and *stevor*. Those families display CVGE, allowing the parasite to switch its antigen during the infection and overcome the immune response. That explains the ability of malaria parasites to maintain chronic infections for a long time [84]. The deeper studied family of CVGs is *var*. This family of around 60 genes encodes for PfEMP1, a transmembrane protein with a hypervariable domain that bind to endothelial surface receptors, determining the parasite tropism [85]. In addition to CVGE, *var* genes present mutually exclusive expression. This means that only one copy of the family is expressed in a single parasite cell, the other copies remaining silenced [86,87].

The mutually exclusive expression implies a complex regulation of activating only one member of the gene family while maintaining the other 59 genes silenced. There are characteristic epigenetic features that accompany the active/silent state. For example, it has been reported that silenced *var* copies are marked with H3K9me3, associated with HP1, and H3K36me3 [76,88]. They appear clustered at the nuclear periphery if located in the subtelomeric ends of the chromosomes, whereas the *var* genes that occupy central positions in the chromosomes cluster via the formation of large chromatin loops [8,79,80,89,90,91]. On the other hand, the active member of the *var* family is marked with H3K9ac and H3K4me3 and the histone variants H2A.Z, H2B.Z and H3.3 [72,89,92,93,94]. The histone variants H2A.Z and H2B.Z are only present in the ring state when the gene is being transcribed. However, in the subsequent trophozoite and schizont stages, the gene remains poised for reactivation in the next cycle. In that case H2A.Z and H2B.Z are not present [15,92,93], but the gene is still marked with the histone variant H3.3 in its promoter region [94].

The transmission of the active state for a particular *var* gene copy, is ensured by H3K4me2, the histone mark responsible of the epigenetic memory in the malaria parasite [95]. In addition to epigenetic marks, ncRNAs have been shown to play an important role in *var* mutually exclusive expression. These are transcribed by the RNA pol II from the conserved introns placed in the *var* genes [96]. An antisense RNA is produced from the intron of the active copy, whereas a sense ncRNA is transcribed from both active and inactive *var* genes. The ncRNA expression is essential for a normal *var* expression, however, their precise mechanism of action is not known [88,97] Accumulating evidence point to the *ruf6* family of lncRNAs playing an important role in *var* mutually exclusive expression [36,98,99]. Comparing ATAC-seq data of two transcriptionally variant *P. falciparum* clonal lines, a recent work revealed that only the *ruf6* genes flanking the active *var* gene appear accessible, whereas there were no differences in promoter or gene body accessibility between active and silent *var* copies. This work proposed that chromatin looping mediated by these *ruf6*-encoding long ncRNA and unknown proteins could participate in *var* gene regulation [36]. Indeed, the RNA pol II itself has been proposed, through its C-terminal domain, to be able to recruit chromatin remodelling factors that erase and rewrite the chromatin state of the selected *var* gene [60,84].

### 3.2. Sexual Differentiation Regulation

Gametocytes are the essential stage of the parasite in transmission of the malaria disease. Becoming gametocytes represents a milestone in the parasite life cycle. A sub-population of the asexually growing parasites commit to stop multiplication and exit intraerythrocytic developmental cycle differentiating into a no-way-back dormant stage until transmitted to the mosquito. This strong link with transmission makes sexual differentiation and gametocytes an attractive target for research and disease control approaches. In *P. falciparum*, usually less than 10% of the parasites produce gametocytes in each multiplication cycle [100]. However, conversion rates are variable between strains [84] due to genetic and non-genetic factors. Indeed, due to the lack of selective pressures, it is common that reference laboratory strains lose the ability to differentiate into sexual forms. This phenotype has been associated to mutations in gametocyte development protein 1 coding gene (*gdv1*) or because deletions of entire chromosomes segments [101]. In nature, however, sexual commitment rates also fluctuate in response to different external and internal factors such as high parasitaemia, vector density (mosquito bites), fever, host nutrition, antimalarial drugs, reticulocyte presence or nutrient restriction [65,102,103]. This evidence suggests that the parasite would be able respond to the within host environment facultatively, adjusting the phenotype, without changes in the genome, to maximize fitness. However, the mechanisms that transduce these host signals to the parasite chromatin are still unknown.

At the transcription level, the main effector in this process has been analysed in several studies, validating the role of AP2-G as the master regulator of sexual differentiation [20,77,104]. This ApiAP2 transcription factor targets a GNGTAC motif that is found in the promoter region of several early gametocyte genes, including its own promoter [21,77]. The *ap2-g* gene is epigenetically silenced by H3K9me3 and HP1 in cycling parasites [23,37,105], while the gametocyte development protein coding gene, *gdv1*, is inhibited by its own antisense RNA [23]. By unknown mechanisms, the *gdv1* gene releases from its antisense RNA and is expressed in stress conditions. Then, GDV1 removes HP1 from *ap2-g* promoter, activating a first wave of its expression [23,106]. Once AP2-G reaches a transcripts peak, its transcription is reduced temporally, stabilizing its expression. However due to its own auto-induction a second expression wave is observed. In this moment, PF3D7_1222400, an adjacent ApiAP2 factor, shows increased levels [21]. The developmental stage in which this AP2-G stabilization occurs determines which of the two possible pathways takes the parasite to sexually differentiate; if it happens in early ring stage, Same Cycle Conversion (SCC) can be taken, starting the differentiation directly [107]. On the contrary, the classical Next Cycle Conversion (NCC) route is taken, where parasites continue its blood development committed until schizogony and reinvasion, starting the differentiation always in the early ring stage. About the environmental signal, it has been proposed that the host-derived lipid lysophosphatidylcholine (LysoPC) induces sexual differentiation through metabolic changes. Its absence would reduce the S-adenosylmethionine (SAM) pool available for epigenetic methylation in order to enhance phosphatidylcholine production, impairing *ap2-g* epigenetic repression [3].

Linked to the sexual differentiation pathway and morphological changes, chromatin organization changes drastically in gametogenesis. An enrichment of H4K20me3, H3K27me3 and H3K36me2 is shown in early gametocytes, indicating a general gene silencing process, while euchromatic marks increase in the late gametocyte stage pointing to a slightly more open chromatin state [9,13] However, HP1 occupancy is higher in all gametocyte stages compared with asexual stages [108]. In late gametocytes, *ap2-g* goes back to the repressive cluster [109]. AP2-G5 binds and silences *ap2-g* as well as early gametocyte genes, being necessary for gametocyte maturation [7]. However, asexual proliferation is not restabilised thanks to AP2-G2, which represses asexual replication genes, being also indispensable for gametocyte maturation [25,110]. Motility genes are the most transcribed in gametocytes, while the less transcribed are genes involved in pathogenesis and invasion. It has been suggested that most of these transcripts, although they are stable, do not translate into proteins and accumulate in gametocytes [56]. This suggests the existence of a transcriptional repression that poise the gene expression [57] in order to prime the gametocyte for mosquito infection.

### 3.3. Erythrocyte Invasion

Different gene families encode for invasion genes allowing alternative pathways for merozoite invasion involving different and specific parasite ligands and erythrocyte receptors. Two of these families, *Pfrh* and *eba*, are formed by mostly nonessential and functionally redundant genes [111,112,113,114]. Those families show Clonally Variant Gene Expression [16,71,73,113,115]. PfRh4 is a member of the PfRh family which is needed for the invasion pathway independent of sialic acid. Its activation allows to change from sialic acid–dependent to a sialic acid–independent invasion pathway [116,117]. This gene is mostly silent but can become active following an epigenetic switch [111,118]. That is, PfRh4 is usually enriched in H3K9me3 and HP1, but when activated, its promoter becomes more accessible, and this is accompanied by a loss of H3K9me3 and a movement of the locus to a nuclear activation site [78,118]. Its expression can be induced when the parasites are with sialic acid-depleted erythrocytes or the main sialic acid–dependent invasion pathway is impaired [111,118,119]. Indeed, the PfRh4 promoter contains putative ApiAP2 motifs [6]. However, the specific transcription factor and its role in opening and activating transcription of the PfRh4 promoters, has to be validated.

### 3.4. Cellular Transport, Nutrients and Drug Resistance

*P. falciparum* can adapt its nutrients uptake and metabolism though epigenetic variation and CVGE. The best described case is the *clag* family. Clag3.1 and Clag3.2 form the plasmodial surface anion channel, which mediates the transport of several nutrients across the erythrocyte membrane [120,121]. *Clag3* family have CVGE, and especially *clag3.1* and *clag3.2* show mutually exclusive expression [16,36,75]. Both genes are localised head-to-tail in the same locus, being one of them accessible and active (marked with H3K9Ac) and the other silenced by heterochromatin (marked with H3K9me3), switching with a low frequency [16,36,59,75,122]. It has been proposed that the alternative *clag3* expression leads to differential nutrient transport, allowing the parasite to adapt to the nutrient environment [61,84,123]. Furthermore, culturing the parasite with the drug blasticidin S results in several changes in the transcriptional state of *clag3.2* and *clag3.1*, pointing to the role of *clag3* and epigenetic factors in controlling nutrient uptake and toxic barring [61,123].

Similarly, members of the fatty acyl-CoA synthases family PfACS can change its expression within the asexual cycle [59] adapting to different nutritional environments [124]. Indeed, the rodent parasite *P. berghei* is able to detect nutrient restriction using the AMPKα homologous kinase KIN and respond through a transcriptional change leading to lower multiplication rates [66]. Similar mechanisms may act in this metabolic adaptation; however, further studies should deepen in it.

### 3.5. Heat Shock Response to Febrile Temperatures

In each intraerythrocytic developmental cycle, following the red blood cell lysis and the release of the merozoites, one of the key symptoms of malaria is fever. As so, *Plasmodium* parasites have evolved mechanisms to adapt and respond quickly to this increase in the host temperature [125]. It has been shown that heat shock affects parasite viability, specially to late stages, and can produce cell cycle arrest [126,127] and gametocyte production [128,129]. As a result of high temperatures, the gene expression profiles change, increasing protein folding genes and stress responsive genes [128,130]. A recent study identified a new ApiAP2 transcription factor as the master regulator of heat shock response: PfAP2-HS [126]. In response to heat, this transcription factor targets the sequence (A/G)NGGGG(C/A) [6], activating the chaperones *hsp70-1* and *hsp90* and an uncharacterized gene PF3D7_1421800 [126]. *Hsp70-1* and *hsp-90* are evolutionary conserved heat response proteins [131,132]. Indeed, a ChIP analysis show some binding of the AP2-HS domain to *hsp70-1* [6]. Previous results also point to a relation between these chaperones and histone deacetylases (HDACs) [133,134]. Two of these epigenetic writers, PfSir2A and PfSir2B, have been shown to be downregulated by Hsp90 in ring stages. This chaperone would recruit H3K9me3 to the promoter regions of PfSir2A and PfSir2B [133], although previous studies show an increment of *sir2* in the same conditions with asynchronized parasite cultures [130]. Despite these exciting results and the potential involvement of AP2-HS in *P. falciparum* adaptive plasticity, the epigenetic basis of AP2-HS mediated heat shock response, the precise mechanism of regulation, and their target genes remain unknown.

## 4. Epitranscriptomics a New Layer in the Malaria Parasite Gene Expression Regulation

Epitranscriptomics is the term that encompasses the different RNA modifications that affect gene expression without changing the primary nucleotide sequence [135]. This ribonucleotide modifications have been proved to play different roles in cell fate and identity in other eukaryotes [136,137]. However, we are just starting to understand the relevance of these new regulatory mechanisms in malaria parasites.

Various tRNA and mRNA epitranscriptomic marks have been identified in *Plasmodium* [58,135,138,139]. A total of 28 of these modifications are associated to tRNA and are developmentally regulated (Figure 2). Most of them reach their peak of abundance in the late stages of the intraerythrocytic developmental cycle and is in those stages where the expression of the proteins displays a codon bias. Some codons are associated to highly expressed genes, while their synonym codons are down represented and overrepresented in the low expressed genes, and vice versa. The codons that are enriched in the upregulated proteins are translated with epitranscriptomally marked tRNAs that are also enriched in late stages, including tRNA^Arg(mcm5UCU)^, tRNA^Gly(mcm5UCC)^, tRNA^Glu(mcm5s2UUC)^, tRNA^Pro(ncm5UGG)^, tRNA^Ser(ncm5UGA)^, tRNA^Val(IAC)^, tRNA^Leu(ncm5UmAA)^, tRNA^Ala(IGC)^, and tRNA^Ile(IAU)^. As a result, the translation efficiency (protein/transcript rate) of the upregulated proteins increases. This effect appears linked to the epitranscriptomic marks in the tRNA anticodon that enhance the recognition of the enriched codons [138]. However, to validate this model individual characterization of each of these modifications would be required.

A tRNA modification in the Cytosine 38 that have been partially characterized is the m5C of the tRNA^Asp(GTC)^. This modification is placed by the methyltransferase PfDNMT2. The PfDNMT2 knockout represses 199 GAC codon biased proteins, and the resulting phenotype displays altered cellular metabolism, protein synthesis and folding. In addition, PfDNMT2 KO parasites are more sensible to nutritional stresses, enhancing the gametocyte production. Without m5C, tRNA^Asp(GTC)^ is degraded under metabolic stress, reducing the expression of the proteins with the GAC codon. However, the RNA nuclease that cleavage m5C-less tRNA^Asp(GTC)^ is not known [135].

On the other hand, mRNA modifications are less studied in *Plasmodium*. Only m7G, pseudouridine (Ψ), m5C, and m6A have been identified in *P. falciparum* [58,139]. Among these, m6A is linked to reduced mRNA stability and low translation efficiency. PfMT-A70 is the methyltransferase responsible of writing this modification. The m6A levels change through the asexual cycle, increasing the modified adenosines in blood late stages (Figure 2). This modification is associated to the motif GGACA and is found surrounding and along the coding sequence [139]. This contrasts with what has been described in other apicomplexan parasites, like *Toxoplasma gondii*, in which the m6A modification is in the 3′ end of the mRNA. In that parasite, this modification has been linked to mRNA maturation and 3′ end polyadenylation [140,141]. Another modification, the mC5 has been associated to transcript stabilisation in gametocytes. This modification is catalysed by NSUN2. Knocking out this protein in the rodent parasite, *P. yoelii* has been shown to impair the gametocytogenesis [58].

All the above epitranscriptomic studies focus on the human blood stages of the parasite and in vitro conditions. It is now fundamental to expand the repertoire of modifications studied, validate their function, and apply the methodology to mosquito stages and in vivo conditions.

## 5. Future Perspectives

Given the importance of epiregulation in the malaria parasite, those mechanisms can be exploited in order to design innovative strategies against the disease. Some advances have been made in this area. Apicidin, a histone deacetylase inhibitor, impairs deeply the transcriptional regulation of the intraerythrocytic developmental cycle via HDAC inhibition. It produces the hyperacetylation of H3K9, H4K8 and the tetra-acetyl H4 (H4Ac4) and demethylation of H3K4me3 in *P. falciparum* [142]. In addition, several histone lysine methyltransferases have been identified. Among them, BIX-01294 and TM2-115 reduce H3K4me3 levels leading to parasite death in asexual stages and gametocytes, their effects have been already tested in mouse models [143,144].

The effect of the epidrugs in gametocytes is key for new malaria strategies that target transmission as this stage is highly resistant to most current malaria treatments [145]. The main problem is that some catalytic domains of the epigenetic writers are well conserved among eukaryotes, opening the possibility to a toxicity in humans [14]. One alternative could be to focus on the differences between the parasite and humans. For example, of the 28 tRNA epitranscriptomic modifications, 3 are present in *P. falciparum* but not in humans (m4Cm, ms2t6A, m3U) [138], so these represent promising targets for future therapies. Another alternative could be to test the epidrugs in the mosquito vector, as it has been already tested with traditional antimalarial drugs [146]. But this possibility has not been assayed yet.

Indeed, the mosquito stages are still one of the greatest unknowns in malaria transcriptional regulation. Most of the studies are conducted in the blood in vitro stages, however, the mosquito host represents an ideal in vivo biological model to analyse gene regulation in the context of host–parasite interactions. Very strong selective forces act in this part of the parasite life cycle, promoting phenotypic variation and plasticity [147,148]. Taking this into account, we assert that the epigenetic and epitranscriptomic regulation of mosquito stages suppose a great seam to unveil the mechanisms of parasite rapid adaptation that might provide new targets for malaria eradication strategies [149].

To conclude, epigenetics represents a fundamental process in *P. falciparum* allowing the malaria parasite to transition and adapt swiftly to the variable and unpredictable environments of its two hosts. The field is now moving towards characterising these regulatory networks in less studied parts of the parasite life cycle, like the mosquito, which are critical to transmission and malaria eradication. On the other hand, the emerging field of epitranscriptomics is revealing the significance of RNA modifications for parasite biology and has the potential to unveil new layers of gene regulation in *Plasmodium* development and adaptation. New weapons against malaria that exploit this new knowledge, like epidrugs that target DNA or RNA modifications, should now be developed.

## Figures and Tables

**Figure 1 genes-13-01734-f001:**
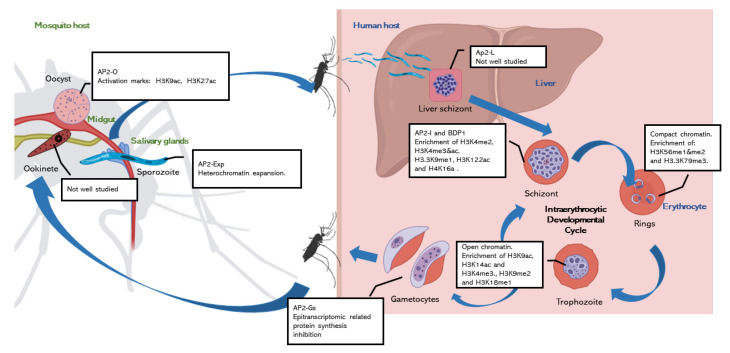
*P. falciparum* life cycle diagram with the more studied life stages at epigenetic level. The white boxes highlight the main epigenetic traits of the corresponding stage. Once the *Anopheles* mosquito bites a human host, sporozoites are released to the bloodstream and migrate to the liver. After hepatocyte invasion, *P. falciparum* invades the red blood cells and starts its Intraerythrocytic Developmental Cycle (IDC). Current evidence shows that in each stage of the blood cycle, chromatin structure is reshaped impacting transcription, and this is associated to differential accessibility, AP2 binding and histone modification patterns. In response to certain within-host factors, some parasite cells exit the IDC and differentiate into the transmissible stage of the gametocyte. The gametogenesis process is under the epigenetic control of the AP2-G regulatory cascade. Parasite transcriptional regulation during development in the mosquito (sporogonic cycle) is little studied, epigenetic data are only available for oocyst and sporozoite mosquito stages. Various AP2 TFs appear to act as master regulators of mosquito-specific parasite genes.

**Figure 2 genes-13-01734-f002:**
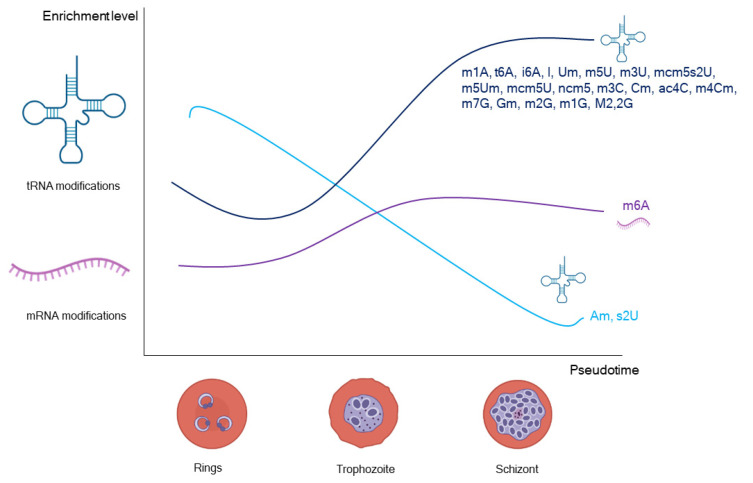
Dynamics of the epitranscriptomic marks through the human blood life cycle. In dark and light blue colors are depicted the tRNA modifications, mRNA modification is showed in purple. The values are not comparable across stages. The graphic is a visual representation of data extracted from Ng et al. 2018 and Baumgarten et al. 2019 [138,139]. Most of the tRNA modifications increase their abundance thorough the life cycle until the trophozoite stage. However, Am and s2U present an opposite pattern, decreasing in abundance in late stages [138]. On the other hand, the mRNA modification m6A is characteristic of late stages [139].

## Data Availability

Not applicable.

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
