# Peer review of "Epigenetic and Epitranscriptomic Gene Regulation in Plasmodium falciparum and How We Can Use It against Malaria"

_genes, 2022, doi:10.3390/genes13101734_

Round 1

Reviewer 1 Report

1. Abstract if very short and some important review should be included here.

2. I saw that several sentences in the background and other parts are lack of citations for example, "Malaria is endemic in 85 countries, being most prevalent in West and Sub-Saharan Africa. The disease is caused by the Apicomplexan parasites of the genus Plasmodium. The Plasmodium parasites infect different mammalian hosts. P. falciparum is the predominant human malaria species, causing the most severe symptomatology and the highest mortality rate". Please try to add citations in the sentences which belong to others.

3. Scienctific name must be italic.

4. The reference styles are different in the manuscript such as " (Comeaux et al., 2011; Crowley et al., 2011; Rovira-Graells et al., 2012, 2015; 353 Ruiz et al., 2018)." and sometimes authors used number citation.

5. Figures are great.

Author Response

We thank the reviewers for their thorough reading of the manuscript and for pointing out several issues that have helped us improved the manuscript in a revised version. Below is a detailed response to each of the points raised by the reviewers.

Reviewer 1

1. Abstract if very short and some important review should be included here

Abstract has been rewritten and expanded to better summarise all the aspects reviewed.

2. I saw that several sentences in the background and other parts are lack of citations for example, "Malaria is endemic in 85 countries, being most prevalent in West and Sub-Saharan Africa. The disease is caused by the Apicomplexan parasites of the genus Plasmodium. The Plasmodium parasites infect different mammalian hosts. P. falciparum is the predominant human malaria species, causing the most severe symptomatology and the highest mortality rate". Please try to add citations in the sentences which belong to others.

We apologise for the lack of citations in some parts. We have revised the text and added citations were appropriate.

3. Scientific name must be italic.

Amended.

4. The reference styles are different in the manuscript such as " (Comeaux et al., 2011; Crowley et al., 2011; Rovira-Graells et al., 2012, 2015; 353 Ruiz et al., 2018)." and sometimes authors used number citation.

This has been fixed, thank you.

5. Figures are great.

Thank you. Figures have been further improved based on comments of reviewer 2.

Reviewer 2 Report

This review article entitled "Epigenetic and epitranscriptomic gene regulation in Plasmodium falciparum and how we can use it against malaria" by Serrano-Durán Rafael, López Farfán Diana and Gómez-Díaz Elena who analyze the epigenetic mechanisms of gene expression regulation mediated mainly by AP2 transcription factors, post-translational histone modifications, and nuclear organization that control cellular plasticity in P. falciparum, as well as epigenetic regulation of RNA expression in parasite development and adaptation. Overall, the text is well written. However, the work requires significant revisions:

1.- The objective of the review is not clear from the abstract.

2.- The Plasmodium genome is AT-rich which differs considerably from other eukaryotes in terms of generating sequence motifs. It would therefore be essential to include information about how this can modify chromatin structure and the binding of transcription factors.

3.- Only 73 transcription factors for 5000 genes have been identified in the Plasmodium falciparum genome. Could you explain how this low number of factors coordinates the gene expression of the more than 5000 genes in the parasite?

4.- Figure 1 should be improved to avoid showing incomplete labelling. Please describe in detail in the figure legend what the figure represents.

5.- Figure 2 is not precise. The figure should stand on its own and not refer to a bibliographic reference to understand its meaning. Explain in more detail what you are trying to represent.

Minor changes:

6.- Write down the authors' addresses.

7.- Please check that reference 4 does not coincide with the text.

8.- Include the reference of the ApiAP2 proteins line 55 and 56.

9.- Please check that Plasmodium is written in italics throughout the text. Also, in the references, it should say Plasmodium falciparum instead of Plasmodium Falciparum. 

Author Response

We thank the reviewers for their thorough reading of the manuscript and for pointing out several issues that have helped us improved the manuscript in a revised version. Below is a detailed response to each of the points raised by the reviewers.

This review article entitled "Epigenetic and epitranscriptomic gene regulation in Plasmodium falciparum and how we can use it against malaria" by Serrano-Durán Rafael, López Farfán Diana and Gómez-Díaz Elena who analyze the epigenetic mechanisms of gene expression regulation mediated mainly by AP2 transcription factors, post-translational histone modifications, and nuclear organization that control cellular plasticity in P. falciparum, as well as epigenetic regulation of RNA expression in parasite development and adaptation. Overall, the text is well written. However, the work requires significant revisions:

1. The objective of the review is not clear from the abstract.

The abstract has been expanded to state more clearly the objective of the review as well as all the different topics covered.

2. The Plasmodium genome is AT-rich which differs considerably from other eukaryotes in terms of generating sequence motifs. It would therefore be essential to include information about how this can modify chromatin structure and the binding of transcription factors.

We thank the reviewer for pointing this out. In the revised version we now comment on the impact of the AT-richness on chromatin structure and transcriptional regulation (page 2, lines 84-90).

3. Only 73 transcription factors for 5000 genes have been identified in the Plasmodium falciparum genome. Could you explain how this low number of factors coordinates the gene expression of the more than 5000 genes in the parasite?

This is indeed a very interesting point. In the revised version we have expanded on the different hypotheses to explain this fact, primarily, sequence redundancy, time-dependent expression and regulatory feed-back loops (page 3, lines 98-105).

4. Figure 1 should be improved to avoid showing incomplete labelling. Please describe in detail in the figure legend what the figure represents.

The figure legend has been completed and we added additional labels to the figure.

5. Figure 2 is not precise. The figure should stand on its own and not refer to a bibliographic reference to understand its meaning. Explain in more detail what you are trying to represent.

We apologise if this figure was not precise. We have added complementary information to the legend so the figure it is now more self-explanatory.

 Minor changes:

6. Write down the authors' addresses.

Author addresses are now included.

7. Please check that reference 4 does not coincide with the text.

This has been corrected, thank you.

8. Include the reference of the ApiAP2 proteins line 55 and 56.

References added.

9. Please check that Plasmodium is written in italics throughout the text. Also, in the references, it should say Plasmodium falciparum instead of Plasmodium Falciparum.

This has been amended.

Round 2

Reviewer 1 Report

Accepted.